DATA RELEASE

# A highly contiguous, scaffold-level nuclear genome assembly for the fever tree (*Cinchona pubescens* Vahl) as a novel resource for Rubiaceae research

Nataly Allasi Canales[1,2,†], Oscar A. Pérez-Escobar[2,3,*,†], Robyn F. Powell[2], Mats Töpel[4], Catherine Kidner[5], Mark Nesbitt[2], Carla Maldonado[6], Christopher J. Barnes[7], Nina Rønsted[1,8], Natalia A. S. Przelomska[2], Ilia J. Leitch[2,*] and Alexandre Antonelli[2,3,9]

1　Natural History Museum of Denmark, University of Copenhagen, Copenhagen, Denmark
2　Royal Botanic Gardens, Kew, Richmond, UK
3　Gothenburg Global Biodiversity Centre, University of Gothenburg, Gothenburg, Sweden
4　University of Gothenburg, Department of Marine Sciences, Gothenburg, Sweden
5　Royal Botanic Garden Edinburgh, Edinburgh, UK
6　Herbario Nacional de Bolivia, Instituto de Ecología, Universidad Mayor de San Andrés, La Paz, Bolivia
7　The Globe Institute, University of Copenhagen, Copenhagen, Denmark
8　National Tropical Botanical Garden, Kalaheo, Hawaii, USA
9　Department of Plant Sciences, University of Oxford, Oxford, UK

**Submitted:** 06 August 2021

\* Corresponding authors. E-mail:
i.leitch@kew.org;
o.perez-escobar@kew.org

† Contributed equally.

Preprint submitted at https://doi.org/10.1101/2022.04.25.489452

## ABSTRACT

The Andean fever tree (*Cinchona* L.; Rubiaceae) is a source of bioactive quinine alkaloids used to treat malaria. *C. pubescens* Vahl is a valuable cash crop within its native range in northwestern South America, however, genomic resources are lacking. Here we provide the first highly contiguous and annotated nuclear and plastid genome assemblies using Oxford Nanopore PromethION-derived long-read and Illumina short-read data. Our nuclear genome assembly comprises 603 scaffolds with a total length of 904 Mbp (~82% of the full genome based on a genome size of 1.1 Gbp/1C). Using a combination of *de novo* and reference-based transcriptome assemblies we annotated 72,305 coding sequences comprising 83% of the BUSCO gene set and 4.6% fragmented sequences. Using additional plastid and nuclear datasets we place *C. pubescens* in the Gentianales order. This first genomic resource for *C. pubescens* opens new research avenues, including the analysis of alkaloid biosynthesis in the fever tree.

**Subjects** Genetics and Genomics, Bioinformatics, Plant Genetics

## DATA DESCRIPTION

### Background

The Andes biodiversity hotspot hosts over 28,000 plant species [1, 2], of which 3,805 are reported to benefit humanity [3]. Unfortunately, nuclear genomic resources are currently available for only 179 species [4]. The fever tree genus (*Cinchona* L., Rubiaceae) comprises 24 species native to the Eastern slopes of the Andean mountain range in South America [5, 6] (Figure 1) and, after coffee, is the second most economically important genus in the

**Figure 1.** Distribution and morphology of *Cinchona pubescens*. (A) *Cinchona* trees in the Andean cloud forest (Photo: O. Pérez-Escobar). (B) The *C. pubescens* specimen studied in this work (CP9014) growing in the Temperate House at the Royal Botanic Gardens, Kew (RBG, Kew), UK (Photo: O. Pérez-Escobar). (C) Flower of *C. pubescens*, CP9014, lateral view (Photo: O. Pérez-Escobar). (D) *Cinchona* barks from the Economic Botany Collection, RBG, Kew, UK (Photo: Nina Rønsted and Nataly Allasi Canales). (E) Distribution map of the *Cinchona* genus across the north-west region of the South American continent shown in blue dots, modified from Maldonado *et al.* [11].

family [7]. The genus is known to be the source of at least 35 quinoline alkaloids, including quinine, which alleviate the fever symptoms associated with malaria [8]. As such, fever trees have been crucial in the economies and livelihoods of local communities for centuries [9, 10].

Despite the historical and economic importance of this genus, DNA sequence datasets for *Cinchona* are limited to only 252 available DNA Sanger sequences [12]. More importantly, there are no nuclear and few reference genomes for any species of the genus. As such, important fundamental and applied questions, such as the mode and tempo of evolution of the fever tree or the genetic pathways responsible for quinine alkaloid production, remain elusive. Previous phylogenetic studies of the Rubiaceae family, specifically of the Cinchonoideae subfamily, which includes the Cinchoneae tribe, are based on a handful of nuclear (Internal Transcribed Spacer of the ribosomal DNA gene [ITS]) and plastid (*mat*K, *rcb*L, *rps*16, *trn*L-F) datasets. They show unresolved relationships between the tribes and

the seven genera of the Cinchoneae tribe that have so far been included in more specific studies [13, 14] (including the genus *Cinchona*, which shows very unclear relationships). Furthermore, studies examining the relationships between species of this genus are equally scarce [9, 10]. A recent genome-wide phylogenetic tree for the order Gentianales [15] provided strong support for *C. pubescens* as a sister to *Isertia hypoleuca* Benth., but the sampling was exclusively at the genus level, so included no other species of *Cinchona* or other genera in the tribe Cinchoneae.

The production of alkaloids is highest in *C. calisaya* Wedd., also known as yellow bark [16, 17]. However, several species in the genus have historically been harvested to provide sources of quinoline alkaloids, one of the most traded natural products, often resulting in significant reductions in their natural ranges and population size [18, 19]. Nevertheless, among them, *C. pubescens*, or red Cinchona bark, is now widely cultivated throughout the tropics, and in some places it has escaped cultivation and become invasive [e.g. in the Galápagos; [20]]. Extensive research on the structure, abundance, and chemical composition of quinoline alkaloids in *Cinchona* [21] has revealed further potential for novel drug discovery. However, the genes involved in the synthetic pathways of quinoline alkaloids, including quinine remain elusive.

## CONTEXT

Nuclear genome assemblies are critical to our understanding of the origin and domestication of useful plants and are a cornerstone resource for breeders [22–24]. Here, we present the first high-quality draft nuclear and plastid genomes of *C. pubescens*, which has a genome size of 1.1 gigabase pairs (Gbp; 1C, this study) and a chromosome number of $2n = 34$. The assemblies were generated using a combination of extensive long-read Oxford Nanopore (~218×) and short-read Illumina paired-end read datasets (~300×) jointly with state-of-the-art genome assemblers. This resulted in a reference genome for which contiguity and quality are comparable to, or even higher [25] than in the five published genome assemblies in Rubiaceae available at the time of submission of this study, namely for *Chiococca alba* (L.) Hitch. [25], *Coffea canephora* Pierre ex A. Froehner [26], *Coffea arabica* L. [27], *Mitragyna speciosa* Korth. [28] and *Neolamarckia cadamba* (Roxb.) Bosser [29]. The plastid genome from short-reads of *C. pubescens* had a length of 156,985 base pairs (bp) and a GC content of 37.74%, very similar to other Rubiaceae plastid genomes [25, 30]. Lastly, we demonstrate the utility and reliability of our resources by constructing nuclear and plastid phylogenomic frameworks of *C. pubescens*.

## METHODS

### Sampling and genomic DNA and RNA sequencing

We sampled leaves from a single *Cinchona pubescens* (NCBI:txid50278) individual, propagated vegetatively from a tree collected in Tanzania in 1977 and cultivated in the Temperate House of the Royal Botanic Gardens, Kew (RBG Kew), UK (accession number 1977-69; a voucher was also prepared, which is deposited in the RBG Kew herbarium [**K**]).

DNA was extracted from fresh tissue using two different protocols to produce paired-end Illumina and native Nanopore libraries. For Illumina DNA library preparation, we used 1000 mg of starting material, which was first frozen in liquid nitrogen then ground using a pestle and mortar. The Qiagen DNeasy (Qiagen, Denmark) plant kit was used to extract DNA from the ground tissue, following the manufacturer's protocol. We built the libraries using



the Illumina TruSeq PCR-free library kit (NEX, Ipswich, MA, USA) following the manufacturer's protocol, by first assessing the DNA quantity and quality using a Nanodrop fluorometer (Thermo Scientific, Denmark) and then fragmenting oligonucleotide strands through ultrasonic oscillation using a Covaris ME220 (Massachusetts, USA) device to yield fragments with an average length of 350 bp. Then, we sequenced the paired-end 150-bp libraries using HiSeq X Ten chemistry (RRID:SCR_016385).

For transcriptome library preparations, total RNA was extracted from 1000 mg of frozen, ground leaf, young bract, mature bract, flower anthesis, flower bud (older), flower bud (young), leaf bud and young leaf tissue using the TRIZoL reagent (Thermo Fisher Scientific, Denmark), following the manufacturer's protocol. Illumina library preparation and sequencing were conducted by Genewiz GmbH (Leipzig, Germany).

Nanopore sequencing data were generated and base-called as part of Oxford Nanopore's London Calling 2019 conference [31]. For Nanopore library preparation, 1000 mg of leaf tissue was frozen and ground with a pestle and mortar. Lysis was conducted using Carlson lysis buffer (100 mM Tris-HCl, pH 9.5, 2% cetyltrimethyl ammonium bromide [CTAB], 1.4 M NaCl, 1% PEG 8000, 20 mM ethylenediaminetetraacetic acid [EDTA]) supplemented with $\beta$-mercaptoethanol. The sample was extracted with chloroform and precipitated with isopropanol. Finally, it was purified using the QIAGEN Blood and Cell Culture DNA Maxi Kit (Qiagen, UK).

Size selection was performed using the Circulomics Short Read Eliminator kit (Circulomics, MD, USA) to deplete fragments below 10 Kbp. DNA libraries were prepared using the Oxford Nanopore Technologies Ligation Sequencing Kit (SQK-LSK109, Oxford Nanopore Technologies, UK). During sequencing on the PromethION platform, reloads were performed when required. Though yield was slightly lower in sequencing for these reloaded samples (over 50 Gbp in 24 hours), the read N50 was over 48 Kbp (up from 28 Kbp without size selection).

## Estimation of genome size

To accurately determine the genome size of *C. pubescens*, we followed the one-step flow cytometry procedure [32], with modifications as described in Pellicer *et al.* [33]. Freshly collected tissue from the same individual sampled for DNA and RNA sequencing was measured together with *Oryza sativa* L. 'IR-36' as the calibration standard (0.49 Gbp/1C [33]) using General Purpose Buffer [34] supplemented with 3% PVP-40 and $\beta$-mercaptoethanol [35]. The samples were analysed on a Partec Cyflow SL3 flow cytometer (Partec GmbH, Münster, Germany) fitted with a 100 mW green solid-state laser (532 nm, Cobolt Samba, Solna, Sweden). Three replicates were prepared and the output histograms were analysed using FlowMax software v.2.4 (Partec GmbH, Münster, Germany). The 1C-value of *C. pubescens* was calculated as 1.1 Gbp, using the equation:

$$1C = (\text{mean peak position of } C. \text{ pubescens}/\text{mean peak position of } O. \text{ sativa}) \times 0.49 \text{ Gbp}$$
$$(= 1C\text{-value of } O. \text{ sativa } [33]).$$

Additionally, using the full Illumina short-read dataset, we implemented a *k*-mer counting method to characterise the genome in Jellyfish v.2.2.10 (RRID:SCR_005491) [36] setting a *k*-mer size of 21. We used GenomeScope (RRID:SCR_017014) to conduct a genome survey and visualise the *k*-mer plot [37]. While we did not deem the *k*-mer counting method

**Table 1.** Summary table for the Illumina whole genomic DNA and RNA-seq libraries, and Oxford Nanopore PromethION corrected reads.

| Biosample accession | SRA accession | Tissue | Source | Technology | Yield (Gb) | Total no. of reads (million) | %GC |
|---|---|---|---|---|---|---|---|
| SAMN30120842 | SRR20784021 | Leaf | Genomic | Illumina HiSeq X | 131 | 428 | 42 |
| SAMN22031859 | SRR16288681 | Bract, leaf, flower | Transcriptomic | Illumina HiSeq X | 115.5 | 385 | 43.5 |
| SAMN30120842 | SRR20784020 | Leaf | Genomic | Oxford Nanopore PromethION | 88.6 | 1.2 | 33.2 |

sufficiently accurate for genome size estimation [38], we used GenomeScope to estimate the genome-wide heterozygosity rates output which was 0.869–0.889%.

## Short read data processing

Sequencing of the DNA Illumina library generated 428 million paired-end reads, representing 131 Gbases (Gb) of raw data. RNA sequencing produced 385 million paired-end reads, representing 115.5 Gb (Table 1). The quality of the raw reads was assessed using FastQC (RRID:SCR_014583) [39], and quality trimming was conducted using the software AdapterRemoval2 v.2.3.1 (RRID:SCR_011834) [40]. Here, bases with a Phred score quality of <30 and read lengths <50 bp were removed together with adapter sequences. The final short-read dataset was 128.4 Gb and comprised 384,626,011 paired reads. This corresponds to an estimated 464.8× coverage (based on the genome size of 1.1 Gbp/1C).

## Plastid genome assembly

The plastid genome of *C. pubescens* was assembled using only the short Illumina reads, as there were some discrepancies using the hybrid dataset. The toolkit GetOrganelle v.1.7.5, was used with the parameters suggested for assembling plastid genomes in Embryophyta (i.e., parameters *-R* 15, *-k* 21,45,65,85,105, *-F* embplant_pt). GetOrganelle produced a single linear representation of the *C. pubescens* plastid genome, with a length of 156,985 bp (Figure 2) and a GC content of 37.74%. These values are very similar to those reported for the *Coffea arabica* plastid genome (155,189 bp in length and 37.4% GC content [30]).

We annotated the plastid genome assembly of *C. pubescens* in CHLOROBOX [41], which implements GeSeq [42], tRNAscan-SE v2.0.5 [43], and ARAGORN v1.2.38 [44]. CHLOROBOX annotations indicated that the *C. pubescens* plastid genome has the typical angiosperm quadripartite structure, i.e., the inverted repeats (IRa and IRb) (each 27,502 bp long), the small single copy region (18,051 bp), and the large single copy region (83,930 bp). We predicted 128 genes, of which 34–37 were tRNA (tRNAScan-SE and ARAGORN, respectively), 81 coding sequences (CDSs), and four ribosomal RNAs (rRNAs). The final structural features of the *C. pubescens* plastid genome were generated using OGDRAW v. 1.3.1 (RRID:SCR_017337) [45] (Figure 2) and edited manually. Finally, the quality of the plastid genome assembly was estimated by mapping the Illumina DNA short reads to the newly assembled genome using the bam pipeline in PALEOMIX v.1.3.6 (RRID:SCR_015057) [46], where we used BWA v.0.7 (RRID:SCR_010910) [47] for alignment, specifying the backtrack algorithm, and filtering minimum quality equal to zero to maximise recovery. After PCR duplicate filtering, the coverage of unique hits was 7960×.

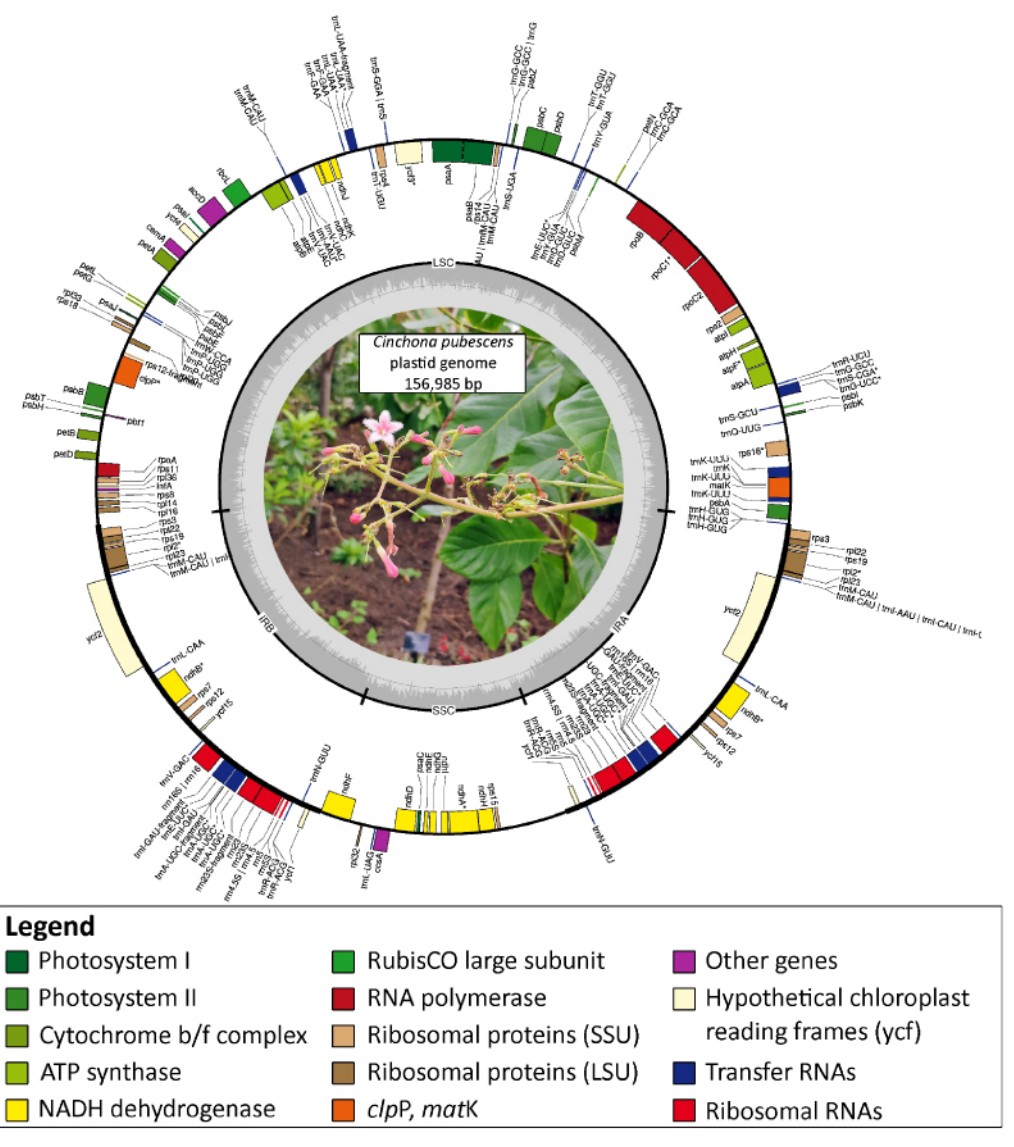

**Figure 2.** Annotated plastome of *Cinchona pubescens*. Genes displayed on the inside of the circle are transcribed clockwise, while genes positioned on the outside are transcribed counter-clockwise.

## Long-read nuclear genome assembly, quality assessment and ploidy level estimation

The quality and quantity of the PromethION sequencing output conducted across four flow cells were evaluated in NanoPlot v.1.82 [48] independently for each flow cell. The sequencing summary report produced by Guppy v3.0.3 was used as input. Overall, the average read length, mean read quality (Phred score) and N50 following base calling with Guppy v3.0.3, and the High Accuracy model reached values of ~19,000 bp, 9, and ~46,000 bp, respectively (Table 2).

A total of 13,252,640 quality-passed reads were produced, representing ~262 Gb and providing a theoretical genome coverage of ~218×. To assemble the raw Nanopore reads



**Table 2.** Summary statistics of the data produced for each PromethION flow cell. The total number of bases and reads (and the % of all reads) produced under different quality thresholds (Q) is also provided.

| Parameter | PAD61766 | PAD61320 | PAD61137 | PAD61315 |
|---|---|---|---|---|
| Active channels | 2,749 | 2,751 | 2,616 | 2,750 |
| Mean read length (bp) | 19,578.4 | 20,249 | 20,067.7 | 19,317.1 |
| Mean read quality (Phred score) | 9.1 | 9.2 | 9.3 | 9.3 |
| Median read length (bp) | 7,033.5 | 8,491 | 8,152 | 7,065 |
| Median read quality (Phred score) | 10.1 | 10.3 | 10.4 | 10.4 |
| Number of reads | 3,603,394 | 3,160,332 | 3,394,747 | 3,094,167 |
| Read length N50 (bp) | 46,786 | 46,891 | 46,496 | 46,467 |
| Total bases | 70,548,548,439 | 63,993,524,635 | 68,124,903,255 | 59,770,483,678 |
| >Q5* | 3,020,566 / (83.8%) / 69,253.6 Mbp | 2,639,509 / (83.5%) / 63,027.1 Mbp | 2,843,776 / (83.8%) / 67,022.5 Mbp | 2,530,628 / (81.8%) / 58,534.5 Mbp |
| >Q7* | 2,719,046 / (75.5%) / 64921.2 Mbp | 2,398,308 / (75.9%) / 59489.4 Mbp | 2,616,120 / (77.1%) / 64036.9 Mbp | 2,307,553 / (74.6%) / 55602.8 Mbp |
| >Q10* | 1,853,358 / (51.4%) / 49964.3 Mbp | 1,698,861 / (53.8%) / 46927.2 Mbp | 1,867,924 / (55.0%) / 51005.6 Mbp | 1,710,969 / (55.3%) / 45629.2 Mbp |
| >Q12* | 545,307 / (15.1%) / 14170.7 Mbp | 548,258 / (17.3%) / 14724.1 Mbp | 624,979 / (18.4%) / 16959.8 Mbp | 708,341 / (22.9%) / 19625.3 Mbp |

*Total number of reads / % of total reads sequenced / Total number of bases.

into scaffolds, we first corrected and trimmed the quality-passed reads using CANU v.1.9 (RRID:SCR_015880) [49] in correction and trimming mode with the following parameters: *genomeSize* = 1.1g, *-nanopore-raw*. This step generated 1,265,511 reads, representing approximately 89 Gb, or a theoretical genome coverage of 74×. Next, the corrected/trimmed reads were used as input into SMARTdenovo v.1.0 (RRID:SCR_017622) [50], using the following parameters: *-c* 1 (generate consensus mode), *-k* 16 (*k*-mer length) and *-J* 5000 (minimum read length). This step produced an assembly comprising 603 scaffolds with an N50 of 2,783,363 bp, representing ~904 Mbp (~82% of the genome size; Table 3). Lastly, a round of scaffold correction was implemented in RACON v.1.4.3 (RRID:SCR_017642) [51] using as input the corrected Nanopore reads generated by CANU and an alignment SAM file produced by mapping the trimmed DNA Illumina reads against the assembly produced by SMARTdenovo. The alignment file was produced by Minimap2 v.2.18 (RRID:SCR_018550) [52] using the "accurate genomic read mapping" settings designed to map short-read Illumina data (flag *-ax*). RACON was executed using an error threshold of 0.3 (*-e* flag), a quality threshold of 10 (*-q*), and a window length of 500 (*-w*). The corrected assembly differed little compared with the raw assembly produced by SMARTdenovo (Table 3).

We followed a two-pronged approach to assess the quality of our corrected nuclear genome assembly. First, we evaluated the proportion of Illumina reads that mapped against our new genome assembly using as input the alignment file (SAM) generated by Minimap2 and computing coverage and mean depth values per scaffold, as implemented in the function *view* (flag *–F* 260) of the software SAMtools v1.12 [53]. Second, we estimated the completeness of the genome as implemented in BUSCO v.5.2 and using viridiplantae_odb10 [54]. A total of 827,098,761 reads were mapped against the corrected genome assembly, representing 99% of the trimmed reads used as input (241,498,983). Mean coverage and read depth ranged from 26–48×. The genome completeness analysis recovered a total of 92.4% conserved eudicot genes, of which 87.5% were single copy, 4.9% duplicated, and 5.6% fragmented. The remaining BUSCO genes were labelled as missing (2%). Taken together, our results suggest that our nuclear genome assembly presents high contiguity and quality with high completeness.

**Table 3.** Summary assembly statistics for *C. pubescens* using SMARTdenovo and RACON software.

| Parameter | SMARTdenovo | RACON |
|---|---|---|
| Size (including Ns) (bp) | 903,037,179 | 9.04E+08 |
| Size (without Ns) (bp) | 903,037,179 | 9.04E+08 |
| Number of scaffolds | 603 | 603 |
| Mean size (bp) | 1,497,574 | 1,499,914 |
| Median size (bp) | 801,066 | 802,662 |
| Longest scaffold (bp) | 14,628,764 | 14,747,124 |
| Shortest scaffold (bp) | 33,171 | 25,882 |
| GC content (%) | 33.17 | 33.07 |
| N50 (bp) | 2,783,363 | 2,802,128 |
| L50 (bp) | 93 | 92 |
| N90 (bp) | 682,446 | 684,435 |
| Gap (%) | 0 | 0 |

Lastly, to evaluate the ploidy level of *C. pubescens* through the newly assembled genome, we computed allele frequencies from reads mapped against two scaffolds, 'utg 230' and 'utg2' derived from our genome assembly, covering 9,568,509 bp (~106× coverage) and 14,628,764 bp (~103× coverage), respectively. The reads were obtained from the mapping procedure conducted to assess the quality of the corrected nuclear genome assembly (see above). We relied on ploidyNGS (RRID:SCR_021320) [55] to compute allele frequencies using the –g option (i.e., guess ploidy levels), a maximum read depth of 100 option (–d 100) and a maximum allele frequency of 0.95 (–m 0.95). Our analysis revealed that the genome of *C. pubescens* is diploid (Figure 3C and D) as inferred by the occurrence of peaks of the frequencies of the two most common alleles close to 50% as well as comparison of Kolmogorov–Smirnoff distances between the allele frequencies computed from our read mappings and those derived from simulated data of known ploidy [56].

## Transcriptome assembly, candidate gene annotation, and quality assessment

To produce a comprehensive database of assembled transcripts, we generated reference-based and *de-novo* assemblies with the Trinity toolkit v.2.8 [56] using trimmed RNA-seq data. The reference-based assembly was conducted using as input the aligned RNA-seq trimmed reads against our new reference genome as produced by aligner STAR v.2.9 (RRID:SCR_004463) [57] with default settings, and a maximum intron length of 57,000 bp as estimated for *Arabidopsis thaliana* (L.) Heynh. (flag *–genome_guided_max_intron*). The *de novo* transcriptome assembly was also produced using the default settings of Trinity and the trimmed RNA-seq reads as input. A comprehensive database of *de novo* and reference-based assembled transcriptomes was compiled using PASA v.2.0.2 (RRID:SCR_014656) [58] and the following parameters: *–min_per_ID* 95, and *–min_per_aligned* 30.

To assess the completeness of the *de novo* transcriptome assembly, we used BUSCO v.5.12 and the representative plant set viridiplantae_odb10, which currently includes 72 species, of which 56 are angiosperms. Our assembled transcriptome captured 92.7% (394/425) of the BUSCO set as complete genes; of the remainder, 3.1% of the genes were fragmented, and 4.2% were missing.

We predicted the structure and identity of the genes in the nuclear genome using the comprehensive transcriptome assembly compiled with PASA. For this purpose, we used

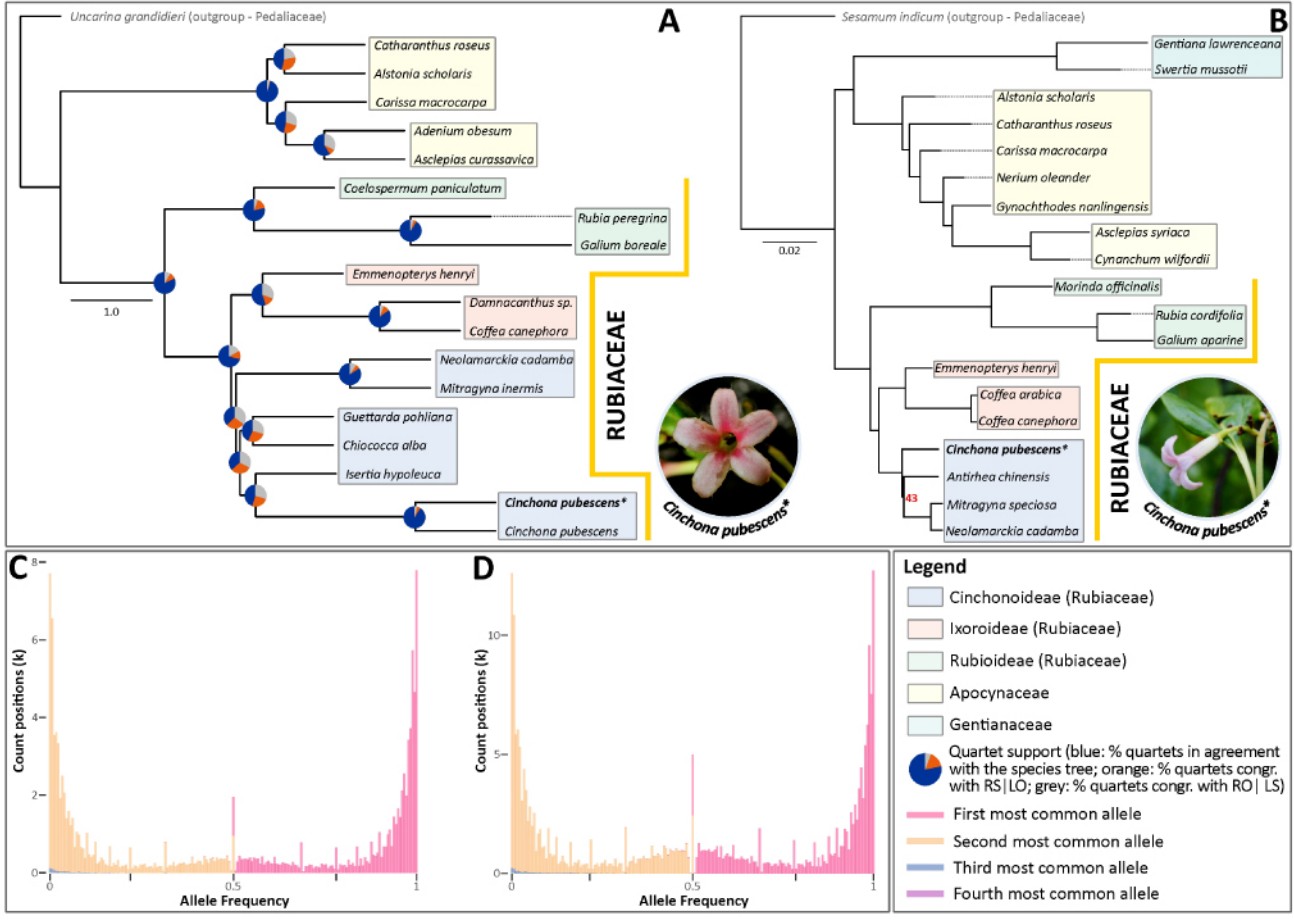

**Figure 3.** The phylogenetic relationships of *Cinchona pubescens* within Gentianales based on nuclear or plastome data together with the ploidy level assessment of the sequenced specimen (CP9014). (A) The coalescent-based species tree estimation of the Gentianales order is inferred from low-copy nuclear gene trees. Pie charts positioned on the nodes represent the percentage of the gene tree quartets agreeing with the topology of the main species tree (blue) and the other two alternative gene tree quartets (orange: second child [R], sister group [S] | first child [L], any other branch [O]; and grey: RO | LS). The genomic data for the *Cinchona pubescens* terminal marked with *, was newly produced in this study. (B) Phylogenetic tree showing relationships between 19 Gentianales species built from whole plastid genome data. Apart from the branch leading to the most recent common ancestor of *Anthirrea, Mitrogyna* and *Neolamarckia*, all the branches from the maximum likelihood tree attained likelihood bootstrap percentages of 100. The coloured boxes indicate the subfamily/family to which sampled taxa belong. (C, D) Ploidy level assessment of *C. pubescens*. On the *x*-axis, allele frequencies for the four most common alleles, derived from reads mapped against the scaffolds 'utg230' (C) and 'utg2' (D). Note that frequency distributions of the first and second most frequent alleles are either skewed towards 0 or 1, denoting homozygote alleles, and 0.5, denoting heterozygote, diploid alleles. (Circular photo insets: Frontal (A) and lateral view (B) of the flower of the sequenced specimen (CP9014) of *C. pubescens* [photos: O. A. Pérez-Escobar]).

AUGUSTUS v.3.3.3 (RRID:SCR_008417) [59] for a combination approach of *ab initio* and transcript evidence-based on RNA-seq data. Since AUGUSTUS considered expressed sequence tags as transcript evidence, we first generated hints from the transcriptome data by aligning the transcripts to the genome using BLAT v.3.5 (RRID:SCR_011919) [60]. Then, we set the hint parameters to rely on the hints and anchor the gene structure. We predicted 72,305 CDSs using the hints and used tomato (*Solanum lycopersicum* L. [Solanales: Solanaceae]) as the reference species. CDS completeness was estimated using BUSCO v.5.12 and viridiplantae_odb10 [53] as reference: 68.4% represented complete BUSCOs, 63.5% were single-copy, 26% were fragmented and 5.6% were missing. We summarized the metrics of the CDS statistics with GenomeQC (Table 4) [61].

**Table 4.** Annotation metrics and summary statistics for the nuclear transcriptome assembly of *C. pubescens*.

| Parameter | Number |
|---|---|
| Number of gene models | 72,305 |
| Minimum gene length (bp) | 275 |
| Maximum gene length (bp) | 65,857 |
| Average gene length (bp) | 4,579.5 |
| Number of exons | 339,698 |
| Average number of exons per gene model | 4.7 |
| Average exon length (bp) | 298.3 |
| Number of transcripts | 72,305 |
| Average number of transcripts per gene model | 1 |
| Number of gene models less than 200 bp in length | 0 |

## DATA VALIDATION AND QUALITY CONTROL

### Nuclear and plastid phylogenomics of *Cinchona*

We verified the phylogenetic placement of the nuclear genome using the reference sequences of the 353 low-copy nuclear genes that are conserved across angiosperms from the Plant and Fungal Trees of Life project [62]. Here, we sampled the gene sequences for 18 taxa from the order Gentianales, including another *C. pubescens* individual from that study [63], that are publicly available in the Tree of Life Explorer [64] hosted by the Royal Botanic Gardens, Kew. To include the data for the *C. pubescens* individual analysed in our study in the analysis of the 353 low-copy nuclear genes of selected Gentianales, we retrieved these genes from our RNA-seq data using the pipeline HybPiper v.1.3.1 [65]. Given the abundance of RNA-seq read data, to render the gene retrieval tractable, as input for HybPiper we used a subsample of the trimmed read data, as implemented in the software seqtk [66]. The gene sequences produced by HybPiper were aligned with the data for the 18 selected Gentianales species using MAFFT v.7.453 [67]; these were then concatenated into a supermatrix for phylogenomic analyses.

We implemented the maximum likelihood approach using RAxML-HPC v.8 (RRID:SCR_006086) [68] with a GTRGAMMA substitution model for each gene and a rapid bootstrap analysis with 500 replications. Then, we filtered the bipartition trees with ≥20% support using Newick utilities [69]. The resulting trees were rooted using phyx v1.2.1 [70], setting *Uncarina grandidieri* (Baill.) Stapf (Lamiales: Pedaliaceae) as the outgroup. To estimate the species tree from the gene trees, we used the coalescent approach with ASTRAL 5.6.1 to calculate the quartet scores, which represent the number of quartet trees present in the gene trees that are also present in the species tree. Q1 shows the support of the gene trees for the main topology, q2 shows the support for the first alternative topology, and q3 shows the support for the second alternative topology [71]. We incorporated these scores into the species tree with an R script [72]. All trees were visualized with FigTree v.1.4.4 (RRID:SCR_008515) [73].

In the nuclear phylogenomic tree derived from the 353 low-copy nuclear genes (Figure 3A), *C. pubescens* clusters within the Cinchonoideae, which is more closely related to the Ixoroideae tribe than to Rubioideae. Most nodes are highly supported by quartet scores, showing that a large proportion of the gene trees agreed with the species tree.

For the plastid phylogeny (Figure 3B), we used *Sesamum indicum* L. (Lamiales: Pedaliaceae) as an outgroup from the Lamiids cluster [74]. We performed maximum likelihood using the complete plastid genomes of the 19 species available to date in the

**Table 5.** Overview of the species and the accession numbers of the specific samples used to infer the phylogenetic tree built using plastid data shown in Figure 3B.

| Taxa | Accession number |
|---|---|
| *Alstonia scholaris* (L.) R. Br. | MG963247 |
| *Antirhea chinensis* Champ. ex Benth. | NC_044102 |
| *Asclepias syriaca* L. | NC_022432 |
| *Carissa macrocarpa* (Eckl.) A.DC. | NC_033354 |
| *Catharanthus roseus* (L.) G. Don. | NC_021423 |
| *Coffea arabica* L. | KY085909 |
| *Coffea canephora* Pierre ex A. Froehner | NC_030053 |
| *Cynanchum wilfordii* (Maxim.) Hook.f. | NC_029459 |
| *Emmenopterys henryi* Oliv. | NC_036300 |
| *Galium aparine* L. | NC_036969 |
| *Gentiana lawrencei* Burkill | KX096882 |
| *Gynochthodes nanlingensis* (Y.Z.Ruan) Razafim. & B.Bremer | NC 028614 |
| *Mitragyna speciosa* Korth. | NC_034698 |
| *Morinda officinalis* F.C.How | NC_028009 |
| *Neolamarckia cadamba* (Roxb.) Bosser | NC_041149 |
| *Nerium oleander* L. | NC_025656 |
| *Rubia cordifolia* L. | NC 047470 |
| *Sesamum indicum* L. (outgroup) | JN637766.2 |
| *Swertia mussotii* Franch. | NC_031155 |

Gentianales (Table 5). All the plastid genomes analysed had the classic quadripartite genomic structure, although some Rubiaceae species have been reported to have a tripartite structure [75]. We aligned the 19 Gentianales plastid genomes with MAFFT v7.427 (RRID:SCR_011811) using the default parameter settings to perform multiple sequence alignments. Then, we estimated the phylogenetic tree with the maximum likelihood approach using the GTRCAT model RAxML-HPC v.8. We conducted heuristic searches with 1000 bootstrap replicates (rapid bootstrapping and search for the best-scoring maximum likelihood tree). Both analyses were performed on the CIPRES Science Gateway [76].

As with the nuclear tree, the plastid trees were also clustered at the subfamily level, recovering the Cinchonoideae, Ixoroideae, and Rubioideae as natural groups, alongside the outgroup species belonging to Pedaliaceae. For the plastid data, most nodes were strongly supported (all but one node attained 100% of likelihood bootstrap support, see Figure 3B). Nevertheless, we found that *Gynochthodes nanlingensis* (Y.Z.Ruan) Razafim. & B.Bremer (Rubioideae) clustered with other Apocynaceae species. While the same result has previously been reported in other studies [77, 78], it seems to be associated with an erroneous DNA sequence attributed to *G. nanlingensis* or a misidentification of the voucher. This requires further verification.

Additionally, the ingroup showed that the Cinchonoideae and Ixoroideae subfamilies are sisters and form a clade, while Rubioideae is placed as sister to this clade. The placement of *C. pubescens* in the Cinchonoideae subfamily cluster using both the plastid and nuclear data presented in this study is consistent with previous taxonomic and phylogenetic studies [79] and supports the robustness of the assembled nuclear and plastid genomes. As potential future work, the raw Nanopore data could be basecalled again using the latest algorithms from Oxford Nanopore to take advantage of recent developments in this area over the last few years, which has seen continuous improvement in raw read accuracy [80, 81].

## REUSE POTENTIAL

Using a combination of extensive short and long-read DNA datasets, we deliver the first highly contiguous and robust nuclear and plastid genome assemblies for one of the historically most traded and economically important *Cinchona* species, *C. pubescens*. The abundant genomic resources provided here open up new research avenues to disentangle the evolutionary history of the Andean fever tree.

In the short term, these genomic tools will significantly help to identify the genes involved in the biochemical pathways synthesizing quinoline alkaloids, identify the underpinning genetic diversity of these genes both between and within species, and open doors on how the expression of these genes is regulated. Our nuclear scaffold-level and plastid genome assemblies will enable future reference-guided assemblies, variant calling, and gene annotation to enhance functional analysis within the *Cinchona* genus, with the potential to further explore the quinine alkaloid biosynthetic pathway in-depth and hence enhance its potential for finding new medicinal leads to treat malaria.

## DATA AVAILABILITY

The genome sequence data, and nuclear and plastid assemblies are available at the NBCI repository, under the BioProject numbers PRJNA768351, PRJNA865567 and PRJNA865558. Supporting data is also available from the GigaDB repository and the low-copy nuclear gene data are available from the Kew Tree of Life Explorer repository (https://treeoflife.kew.org/tree-of-life) [62].

## DECLARATIONS

### List of abbreviations

Bp: base pair; CDS: coding sequence; Gb: gigabase; Kbp: kilobase pair.

### Ethical approval

Not applicable.

### Consent for publication

Not applicable.

### Competing Interests

The authors declare that they have no competing interests.

### Funding

NR, AA, CB, and NAC received funding from H2020 MSCA-ITN-ETN Plant.ID, a European Union Horizon 2020 research and innovation programme under grant agreement No 765000. OAPE acknowledges financial support from the Swiss Orchid Foundation and the Lady Sainsbury Fellowship at the Royal Botanic Gardens, Kew. PromethION sequencing (flow cells and consumables) were provided by Oxford Nanopore Technologies. AA and NR acknowledge funding from the SciLifeLab 2015 Biodiversity Program. AA is further funded by the Swedish Research Council and the Royal Botanic Gardens, Kew.

## Authors' contributions

IJL, OAPE, NR, MT and AA conceived the study. OAPE, AA and MN collected plant tissue. OAPE, IJL, RFP, MT and AA generated datasets. OAPE, NAC, CK, NASP and MT conducted in-silico analyses. OAPE, NAC and CM prepared the figures. NAC and OAPE wrote the manuscript, with contributions from all co-authors. All authors read and approved the final version of the manuscript.

## Acknowledgements

We thank Jonathan Pugh, Vania Costa, and Simon Mayes for support and assistance during Nanopore sequencing preparation and Claes Persson for taxonomic advice. We also thank the two anonymous reviewers and the associate editor who provided constructive feedback to this manuscript.

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
