## [Reviewer Report]

Comments on revised manuscriptI’ve read the author’s responses to all the reviewer comments and read the updated manuscript and I am satisfied with the changes made.

---

## [Reviewer Report]

Reviewer name and names of any other individual's who aided in reviewer John HamiltonDo you understand and agree to our policy of having open and named reviews, and having your review included with the published papers. (If no, please inform the editor that you cannot review this manuscript.)YesIs the language of sufficient quality?YesPlease add additional comments on language quality to clarify if needed
Are all data available and do they match the descriptions in the paper? YesAdditional Commentsyes, downloaded and checked from the Gigabyte FTP siteAre the data and metadata consistent with relevant minimum information or reporting standards? See GigaDB checklists for examples <a href="http://gigadb.org/site/guide" target="_blank">http://gigadb.org/site/guide</a>YesAdditional CommentsI was unable to check sequence data deposited in the SRAIs the data acquisition clear, complete and methodologically sound?YesAdditional CommentsYes, but summary tables are missing for the Illumina WGS and RNA-Seq sequencing in the manuscriptIs there sufficient detail in the methods and data-processing steps to allow reproduction?NoAdditional CommentsSome parts of the manuscript are very good in this respect and some parts (esp, annotation) are missing parameters and core details.Is there sufficient data validation and statistical analyses of data quality? NoAdditional CommentsEspecially the analyzing of the quality /completeness of the genome annotation.Is the validation suitable for this type of data?YesAdditional CommentsWhere it is not missing, it is suitableIs there sufficient information for others to reuse this dataset or integrate it with other data?YesAdditional CommentsAny Additional Overall Comments to the AuthorIn this manuscript, Canales et al. present the long-read based assembly and annotation of the genome of fever tree (Cinchona pubescens), well known as the source of quinine alkaloids traditionally used to treat malaria. This will be a genome of interest and welcome resource for the community. I enjoyed reading this manuscript about this interesting species and I have several comments:  1. There is not a summary table for the Illumina WGS and the three RNA-Seq libraries. This should be added.  2. Since you have Illumina WGS short reads, it would be informative to add a Genomescope kmer plot (http://qb.cshl.edu/genomescope/) as an additional estimate of genome size and heterozygosity to section 1.3  3. The BUSCO metrics for the assembly are lower than expected. I believe this is due the lack of sufficient genome polishing. Refer to the Solanum pennellii genome paper (https://doi.org/10.1105/tpc.17.00521) where they used a similar assembly strategy and discuss the need for adequate polishing (see “Prior to Polishing, Genome Error Rate Is Substantial”).   4. Section 1.6 – It is noted that PASA describes transcript evidence as ESTs which is a legacy from the time it was developed, but then the RNA-seq transcript assemblies are also described as ESTs later in the section which is incorrect and confusing.   5. There is not an assessment of the annotation, just a statement of the number of CDSs predicted. This is an issue as the number of CDSs is far higher than reported in related species. There is not a discussion of repeat masking the genome assembly so I am assuming AUGUSTUS was run on the unmasked assembly with no downstream filtering or refinement. Doing this increases the number of TE-related gene models and annotation artifacts. As this is a data note/data release there should really be at a minimum: a. A table summarizing the annotation in the manuscript b. An analysis to identify models with evidence support c. BUSCO results for the annotation
RecommendationMajor Revision

---

## [Reviewer Report]

Reviewer name and names of any other individual's who aided in reviewer Bing bing LiuDo you understand and agree to our policy of having open and named reviews, and having your review included with the published papers. (If no, please inform the editor that you cannot review this manuscript.)YesIs the language of sufficient quality?YesPlease add additional comments on language quality to clarify if needed
Are all data available and do they match the descriptions in the paper? YesAdditional CommentsAre the data and metadata consistent with relevant minimum information or reporting standards? See GigaDB checklists for examples <a href="http://gigadb.org/site/guide" target="_blank">http://gigadb.org/site/guide</a>YesAdditional CommentsIs the data acquisition clear, complete and methodologically sound?YesAdditional CommentsIs there sufficient detail in the methods and data-processing steps to allow reproduction?YesAdditional CommentsIs there sufficient data validation and statistical analyses of data quality? YesAdditional CommentsIs the validation suitable for this type of data?YesAdditional CommentsIs there sufficient information for others to reuse this dataset or integrate it with other data?YesAdditional CommentsAny Additional Overall Comments to the AuthorI was very pleased to read your article on Fever tree's genomes, and I think it is a very valuable foundational work. The assembled genome recovered ~85% (903M or 904M, table1) of the estimated genome size (1.1 Gb/1C) with an N50 = 2802128 bp; 72,305 CDSs were annotated and 83% (or 87.6%, line 207) of BUSCOs were recovered, but there is a lack of clarity around these statistics in the study. And it is necessary to provide the repeat annotations, function annotations and non-coding RNA annotations. Besides, the BUSCOs recovered is no more than 90%, you should give your explanation.   Minor comments Lines 34-43, you should add the plastid genome results here. Lines 38, check the genome size. Maybe 904M? Lines 41 and 207, you gave two different percentages of BUSCOs, please check. Lines 144,145 and 149, the numbers of reads and bases are non-correspondence, please check, as the read length is 150bp. Lines 172, I doubt about the overall mapping re (7.34%). Lines 198, you should add the description about the genome produced by RACON. Lines 204 and 223, why do you use the different version of BUSCO software? Lines 206-207, why you did not give the result of mapping rio. Lines 243, you should provide the BUSCO result of proteins (CDSs). Lines 257 and 280, why you use different version of MAFFT? Lines 289, check the sentence ‘had a BS of 100%’.
RecommendationReject (Unsound or Unusuable)